# Low-Intensity Resistance Training with Moderate Blood Flow Restriction Appears Safe and Increases Skeletal Muscle Strength and Size in Cardiovascular Surgery Patients: A Pilot Study

**DOI:** 10.3390/jcm10030547

**Published:** 2021-02-02

**Authors:** Hironaga Ogawa, Toshiaki Nakajima, Ikuko Shibasaki, Takahisa Nasuno, Hiroyuki Kaneda, Satoshi Katayanagi, Hayato Ishizaka, Yuta Mizushima, Azusa Uematsu, Tomohiro Yasuda, Hiroshi Yagi, Shigeru Toyoda, Tibor Hortobágyi, Takashi Mizushima, Teruo Inoue, Hirotsugu Fukuda

**Affiliations:** 1Department of Cardiovascular Surgery, School of Medicine, Dokkyo Medical University, Shimotsuga-gun, Tochigi 321-0293, Japan; hironaga_0722@yahoo.co.jp (H.O.); sibasaki@dokkyomed.ac.jp (I.S.); fukuda-h@dokkyomed.ac.jp (H.F.); 2Department of Cardiovascular Medicine, School of Medicine, Dokkyo Medical University, Shimotsuga-gun, Tochigi 321-0293, Japan; tnasuno@dokkyomed.ac.jp (T.N.); hirokane1010@gmail.com (H.K.); hyagi881@mac.com (H.Y.); s-toyoda@dokkyomed.ac.jp (S.T.); inouet@dokkyomed.ac.jp (T.I.); 3Department of Medical KAATSU Training, Dokkyo Medical University, Shimotsuga-gun, Tochigi 321-0293, Japan; 4Department of Rehabilitation, Dokkyo Medical University Hospital, Shimotsuga-gun, Tochigi 321-0293, Japan; kata-s@dokkyomed.ac.jp (S.K.); i-hayato@dokkyomed.ac.jp (H.I.); yu-mizu@dokkyomed.ac.jp (Y.M.); mizusima@dokkyomed.ac.jp (T.M.); 5Department of Health and Sport Sciences, Premedical Sciences, Dokkyo Medical University, Shimotsuga-gun, Tochigi 321-0293, Japan; auematsu@dokkyomed.ac.jp; 6School of Nursing, Seirei Christopher University, Hamamatsu, Shizuoka 433-8558, Japan; tomohiro-y@seirei.ac.jp; 7University Medical Center Groningen, University of Groningen, Groningen, 9713 GZ Groningen, The Netherlands; t.hortobagyi@umcg.nl

**Keywords:** cardiac surgery, resistance exercise, muscle hypertrophy, cardiac rehabilitation, moderate blood flow restriction, KAATSU training, sarcopenia

## Abstract

We examined the safety and the effects of low-intensity resistance training (RT) with moderate blood flow restriction (KAATSU RT) on muscle strength and size in patients early after cardiac surgery. Cardiac patients (age 69.6 ± 12.6 years, *n* = 21, M = 18) were randomly assigned to the control (*n* = 10) and the KAATSU RT group (*n* = 11). All patients had received a standard aerobic cardiac rehabilitation program. The KAATSU RT group additionally executed low-intensity leg extension and leg press exercises with moderate blood flow restriction twice a week for 3 months. RT-intensity and volume were increased gradually. We evaluated the anterior mid-thigh thickness (MTH), skeletal muscle mass index (SMI), handgrip strength, knee extensor strength, and walking speed at baseline, 5–7 days after cardiac surgery, and after 3 months. A physician monitored the electrocardiogram, rate of perceived exertion, and the color of the lower limbs during KAATSU RT. Creatine phosphokinase (CPK) and D-dimer were measured at baseline and after 3 months. There were no side effects during KAATSU RT. CPK and D-dimer were normal after 3 months. MTH, SMI, walking speed, and knee extensor strength increased after 3 months with KAATSU RT compared with baseline. Relatively low vs. high physical functioning patients tended to increase physical function more after 3 months with KAATSU RT. Low-intensity KAATSU RT as an adjuvant to standard cardiac rehabilitation can safely increase skeletal muscle strength and size in cardiovascular surgery patients.

## 1. Introduction

Cardiovascular patients undergoing cardiovascular surgery are often physically frail and have low muscle mass, i.e., sarcopenia [1,2]. Preoperative frailty and sarcopenia predict elderly patients’ post-operative medical status [3,4]. These patients tend to remain weak and suffer from muscle atrophy after cardiac surgery [5]. Therefore, cardiac rehabilitation aims to reduce the loss of muscle strength and size and increase cardiopulmonary function so that quality of life is maintained as much as possible after open-heart surgeries.

Traditionally, cardiac rehabilitation consists of below-anaerobic threshold, low-intensity aerobic exercise [6,7], which can improve exercise capacity but has only minor effects on skeletal muscle strength and mass [8]. Patients with frailty and sarcopenia especially need to increase muscle strength and mass but low-intensity aerobic exercise alone is ineffective for this purpose. Therefore, cardiac rehabilitation should include not only aerobic but also resistance training (RT) [9,10]. According to the American College of Sports Medicine (ACSM) [11], RT at 60~70% of one repetition maximum (1-RM) delivered in 3–4 sets of 8–12 repetitions to exhaustion, is optimal to improve maximal voluntary force and induce muscle hypertrophy. However, early after cardiac surgery, elderly patients are often unable to perform high-intensity RT. A safe and effective version of RT is needed that can still improve muscle strength and size in patients early after cardiac surgery.

The novel moderate blood flow restriction (KAATSU) training moderately restricts blood flow by compressing the proximal portion of the lower or upper extremities with a specially-designed cuff. It is a well-established method to increase muscle strength and size in athletes and healthy subjects through short-term low-intensity RT (20–30% of 1-RM) [12,13,14,15,16]. The mechanisms by which KAATSU training potentiates the effects of low-intensity RT remain unclear, but appear to be related, in part, to an increase in muscle activation [12,17,18,19]. Indeed, KAATSU RT can enhance muscle activation during low-intensity knee extension exercise in cardiovascular patients with or without cardiovascular surgery [20]. Previous studies reported that KAATSU RT was accompanied by potential adverse side effects such as dizziness, subcutaneous hemorrhage, petechial hemorrhage, drowsiness, numbness, nausea, itchiness, etc. [21,22,23]. Until now, KAATSU RT has been used without adverse side effects in the physical rehabilitation of older adults [16], in patients recovering from an anterior cruciate ligament (ACL) surgery [24], and in patients with ischemic heart disease [25]. Thus, KAATSU RT has the potential to be an effective method to improve muscle strength and size in patients with cardiac surgery. However, it remains unexplored if low-intensity KAATSU RT can safely increase muscle strength and size in patients early after cardiac surgery. The purpose of the present study was to determine if low-intensity KAATSU RT can safely increase muscle strength and size in patients undergoing cardiac open surgery. We hypothesized that low-intensity KAATSU RT also provides beneficial effects in cardiovascular surgery patients.

## 2. Materials and Methods

### 2.1. Patients

Of the inpatients receiving cardiac open surgery between April 2017 and June 2020, a total of 25 patients, who met the following criteria (Table 1), were recruited to participate the present study (Figure 1). Patients’ surgeons determined if a patient met the inclusion criteria. Of these, 21 patients (69.6 ± 12.6 years, range: 23–89, *n* = 21, M = 18), who agreed with the study, participated in the research. Patients underwent coronary artery bypass grafting (CABG, *n* = 1), Bentall operation (*n* = 1), aortic valve replacement (AVR, *n* = 11), mitral valve replacement (MVR, *n* = 2), or mitral valve plasty (MVP, *n* = 6). We also assessed the incidence of conventional risk factors such as fibrillation. Eighteen patients had New York Heart Association (NYHA) class II, and 3 patients had NYHA class III (Table 2).

Patients were randomly assigned either to a control group (*n* = 10) or a KAATSU RT group (*n* = 11) (Figure 1). Table 2 summarizes patients’ baseline characteristics. All patients had received the inpatients’ standard cardiac rehabilitation program consisting of 30 min aerobic exercise within the level of anerobic threshold (AT) on a cycle ergometer. After discharge, patients in the control group performed the standard cardiac rehabilitation program in the hospital twice a week for 3 months but did not receive KAATSU RT. Patients in KAATSU RT group participated in the KAATSU RT program twice a week, adding on standard cardiac rehabilitation in the hospital for 3 months. The Regional Ethics Committee approved the study protocol (approval number: 27074), which was conducted according to Declaration of Helsinki. Each patient provided a written consent. This pilot study was registered at ISRCTN (Registration code ISRCTN11180246).

### 2.2. Reduction of Femoral Muscle Blood Flow by KAATSU

We used previously detailed methods of blood flow restriction [12,13,18,26,27,28]. The compact KAATSU apparatus (KAATSU NANO, KAATSU JAPAN Co., Ltd., Tokyo, Japan) was used for KAATSU RT and a pressure cuff was attached to the base of each thigh. A specially-designed KAATSU cuff (KAATSU NANO, KAATSU JAPAN Co., Ltd.) applied pressure at the proximal end on each thigh to moderately restrict muscle blood flow. The cuff pressure was first set at 100 mmHg at a mounting pressure of 20 mmHg and was gradually increased to 160–200 mmHg over 2 to 3 weeks, depending on patients’ Borg rate of perceived exertion (RPE) [29] during the KAATSU RT.

### 2.3. KAATSU RT Protocol

RT started 5–7 days after surgery if patients in KAATSU RT group were able to walk 200 m. RT included seated knee extension and flexion and leg press exercises with moderate blood flow restriction twice a week for 3 months. A medical doctor supervised the RT program by continuously monitoring vital signs, electrocardiogram, RPE, and the color of lower extremities as recommended by guidelines for the rehabilitation of patients with cardiovascular disease [7].

With blood flow moderately restricted, patients in KAATSU RT group performed knee extension (GX-320, OG wellness company, Okayama, Japan) followed by leg press exercise (GX-310, OG wellness company, Okayama, Japan) on exercise machines. Paced by a metronome, the duration of the specific phase (eccentric and concentric) was 1.5 s for both exercises. To minimize the risk of an adverse event, patients started to exercise at a very low exercise intensity (5–10 kg for leg extension, 20–30 kg for leg press) and very low exercise volume (a single set consisted of 20 repetitions). When patients’ RPE was below 15 after exercise, the number of repetitions was increased to 30 and then the number of sets increased to 3 in a stepwise manner, reaching 3 sets of 30 repetitions for each exercise with 30 s of inter-set rest. After week 2 of RT, we estimated KAATSU RT group patients’ 1-RM of each exercise by evaluating their 8 or 10-RM. Exercise intensity was then adjusted to 20–30% of estimated 1-RM. We carefully controlled exercise intensity and volume to keep post-exercise RPE below 15.

### 2.4. Evaluation of Clinical Data

Each patient received a preoperative transthoracic echocardiography. Two-dimensional images were recorded with an iE33 and EPICQ7 cardiovascular ultrasound system (PHILIP, Amsterdam, The Netherlands). Left ventricular ejection fraction (EF) was calculated using the Simpson method. We evaluated the following clinical data at baseline (2–3 days before surgery), early post-surgery (5–7 days after cardiac surgery when patients were able to walk 200 m), and approximately at 3 months after surgery in both groups: (1) blood sampling and biochemical analysis; (2) physical function (handgrip strength, isometric knee extension peak torque, walking speed), and (3) bioelectrical impedance analysis (BIA) and anterior mid-thigh wall thickness (MTH) using B-mode transverse ultrasound imaging.

(1).Blood biochemistry: Fasting venous blood samples were obtained from the antecubital vein and analyzed for: hemoglobin A1 (HbA1c), albumin (Alb) brain natriuretic peptide (BNP), creatinine, creatine phosphokinase (CPK), high-sensitivity C-reactive protein (hsCRP), prothrombin time-international normalized ratio (PT-INR), and D-dimer (only for KAATSU RT group) using routine biochemical analysis performed in the hospital’s clinical laboratory.(2).Functional assessment: Maximum voluntary isometric contraction (MVIC) of the handgrip was determined with a factory-calibrated handgrip dynamometer (TKK 5401, TAKEI Scientific Instruments Co., Ltd., Tokyo, Japan). Each patient completed two trials and the higher value was used in the analysis. MVIC of the knee extensors was determined with a digital handheld dynamometer (μTas MT-1, ANIMA Co., Ltd., Tokyo, Japan) as described previously [1,30,31]. Patients were seated in a chair with the trunk vertical, the hip and knee joints flexed 90°, and arms folded across the chest. The load cell was affixed with a belt on the anterior aspect of the tibia just above the ankle so that the force was applied to the loadcell at 90°. Each patient completed two trials with 2 min of inter-trial rest. The higher of the two values was used in the analysis. Walking speed was computed as the time needed to walk 4 m at a habitual pace.(3).Body composition: A multi-frequency bioelectrical impedance analyzer (BIA) (InBody S10 Biospace devise, Biospace Co., Ltd., Korea/Model JMW140) was used according to the manufacturer’s recommendations [1,31]. Thirty impedance measurements were obtained at 1, 5, 50, 250, 500, and 1000 kHz over the right and left arms and legs and the trunk. The measurements were carried out while the subjects rested quietly in the supine position, with elbows extended and relaxed along the trunk. BIA-derived body composition metrics included body fat volume, % body fat, extracellular water (ECW), and total body water (TBW), and the ECW/TBW ratio was computed. Skeletal muscle mass index (SMI; appendicular skeletal muscle mass/height^2^, kg/m^2^) was measured as the sum of lean soft tissue of the two upper limbs and the two lower limbs. In this study, sarcopenia was defined according to the Asian Working Group for Sarcopenia (AWGS) [32] criteria (handgrip strength < 28 kgf for males and <18 kgf for females or walking speed ≤ 1.0 m/s; SMI < 7.0 kg/m^2^ for males and <5.7 kg/m^2^ for females).(4).Muscle size: Muscle thickness of thigh (MTH) was measured at the midpoint of the thigh length (between the greater trochanter and the lateral femoral condyle) with a real-time linear electronic scanner using a 10 MHz scanning head and ultrasound probe (L4-12t-RS Probe, GE Healthcare, Tokyo, Japan) and ultrasound (LOGIQ e, GE Healthcare, Tokyo, Japan) [1,30,31,33]. The scanning head was coated with a water-soluble transmission gel to provide acoustic contact without depressing the dermal surface. The subcutaneous adipose tissue-muscle interface and the muscle-bone interface were identified from the image. The perpendicular distance from the adipose tissue-muscle interface to muscle-bone interface was considered to represent MTH, measured in the supine position. Ink markers on the anterior and posterior thigh and posterior lower leg were used to ensure similar positioning over repeated MTH measurements. The measurement was performed twice on the right thigh and the average of the two values were used in the analysis. The test-retest reliability (Intraclass Correlation Coefficient (ICC), standard error of measurement (SEM), and minimal difference) was previously determined using the data of 9 older women measured twice within a few days for MTH (0.994, 0.28 cm, 0.79 cm).(5).Side effects: To verify the safety of KAATSU RT, all adverse and serious adverse events including deterioration of circulatory hemodynamics and hospitalizations were carefully monitored and recorded. We monitored the following potential side effects reported previously in conjunction with KAATSU RT: dizziness, subcutaneous hemorrhage, petechial hemorrhage, drowsiness, numbness, nausea, itchiness, etc. [21,22,23]. Furthermore, because surgery is a risk for deep vein thrombosis and KAATSU RT may increase the risk for developing new deep vein thrombosis in cardiac surgery patients, we carefully monitored patients’ lower limbs during the KAATSU RT. We also evaluated CPK and D-dimer after 3 months.

### 2.5. Statistical Analysis

Data are presented as mean ± SD. The main analysis for the parametric data was a 2-way ANOVA (group: control and KAATSU RT, time: baseline, early after cardiac surgery, after 3 months) and Bonferroni’s multiple comparison was used for post hoc test. For the non-parametric data, Mann–Whitney U-test or chi-square test was main analysis. We computed Pearson’s correlation coefficient to determine if baseline muscle size and function would be associated with changes in muscle size and function after 3 months. A *p*-value less than 0.05 was set as significant level. The *p*-value was corrected with Bonferroni’s method. All analyses were performed with the software (SPSS version 24 for Windows, IBM Corp., NY, USA).

## 3. Results

### 3.1. Baseline Characteristics of the Patients

Before cardiac surgery, there were no differences between the two groups in age (*p* = 0.08, *d* = 0.82), body mass (*p* = 0.80, *d* = 0.11), body mass index (BMI) (*p* = 0.45, *d* = 0.31), and height (*p* = 0.34, *d* = 0.19). There were no differences in risk factors between the two groups (all *p* > 0.05, Table 2). The preoperative creatinine (*p* = 0.25, *d* = 0.54), Hb (*p* = 0.51, *d* = 0.30), Alb (*p* = 0.06, *d* = 0.89), HbA1c (*p* = 0.33, *d* = 0.44), BNP (*p* = 0.14, *d* = 0.69), and EF (*p* = 0.31, *d* = 0.45) also did not differ between the two groups (Table 2).

Table 3 shows the morphological and physical function data. At baseline, there were no differences between the two groups in body fat volume (*p* = 0.63, *d* = 0.21), body fat percentage (*p* = 0.90, *d* = 0.05), ECW/TBW (*p* = 0.19, *d* = 0.62), MTH (*p* = 0.30, *d* = 0.46), SMI (*p* = 0.77, *d* = 0.13), walking speed (*p* = 0.72, *d* = 0.16), handgrip strength (*p* = 0.76, *d* = 0.14), and knee extensor strength (*p* = 0.54, *d* = 0.27).

### 3.2. Body Composition

There was a group by time interaction in body mass (F = 4.8, *p* = 0.016). Compared with baseline, body mass did not differ early after cardiac surgery (*p* = 0.061, *d* = 0.88, −2.1 kg, −3.3%) and after 3 months in the control group (*p* = 0.11, *d* = 0.78, −1.3 kg, −2.1%). Compared with baseline, body mass decreased early after cardiac surgery in the KAATSU RT group (*p* = 0.003, *d* = 1.35, −3.0 kg, −4.9%), and then returned after 3 months to the level at baseline (*p* = 1.0, *d* = 0.24, +0.7 kg, +1.1%).

There were no groups by time interaction but the time main effect in BMI (F = 4.7, *p* = 0.015) and body fat volume (F = 4.6, *p* = 0.016).

### 3.3. Muscle Size

There was a group by time interaction in MTH (F = 12.0, *p* = 0.0001) and SMI (F = 3.8, *p* = 0.03). In the control group, compared with baseline, muscle size did not differ early after cardiac surgery (MTH: *p* = 0.07, *d* = 0.88, −0.2 cm, −7.1%, SMI: *p* = 0.16, *d* = 0.70, −0.3 kg/m^2^, −4.3%), and after 3 months (both MTH and SMI: *p* = 1.0, *d* < 0.01). In the KAATSU RT group, compared with baseline, muscle size decreased early after cardiac surgery (MTH: *p* = 0.01, *d* = 1.21, −0.4 cm, −16.0%, SMI: *p* = 0.04, *d* = 0.92, −0.4 kg/m^2^, −5.8%), and then increased after 3 months to the level above baseline (MTH: *p* = 0.0001, *d* = 1.85, +0.5 cm, +20.0%, SMI: *p* = 0.02, *d* = 1.05, +0.4 kg/m^2^, +5.8%).

### 3.4. Physical Function

There was a group by time interaction in walking speed (F = 5.1, *p* = 0.01). Compared with baseline, walking speed did not differ early after cardiac surgery (*p* = 0.69, *d* = 0.41, −0.1 m/s, −9.1%) and after 3 months (*p* = 1.0, *d* = 0.14, +0.1 m/s, +9.1%) in the control group. Compared with baseline, walking speed did not differ early after cardiac surgery (*p* = 0.31, *d* = 0.54, +0.1 m/s, +9.1%) but increased substantially more after 3 months to a level above baseline (*p* = 0.001, d = 1.9, +0.29 m/s, +26.4%) in the KAATSU RT group.

There was no group by time interaction but the time main effect in handgrip strength (F = 5.7, *p* = 0.007).

There was a group by time interaction in knee extensor strength (F = 7.4, *p* = 0.002). Compared with baseline, knee extensor strength did not differ early after cardiac surgery (*p* = 0.17, *d* = 0.70, −5.5 kgf, −16.4%) and after 3 months in the control group (*p* = 0.57, *d* = 0.45, −1.8 kgf, −5.4%). Compared with baseline, knee extensor strength did not differ early after cardiac surgery (*p* = 1.0, *d* = 0.16, −1.3 kgf, −4.3%) and then increased after 3 months to a level above baseline (*p* = 0.006, *d* = 1.24, +11.3 kgf, +37.0%) in the KAATSU RT group.

### 3.5. Correlation Analyses for Muscle Size and Function in the Control and KAATSU RT Group

Figure 2 shows the relationship between baseline and changes after 3 months in MTH, SMI, walking speed, and knee extensor strength. 

Changes in MTH after 3 months did not correlate with baseline MTH at baseline in the control (r = −0.09, *p* = 0.814), but this relationship was moderate and negative in the KAATSU RT group (r = −0.57, *p* = 0.07).

Changes in SMI after 3 months did not correlate with SMI at baseline in the control (r = −0.17, *p* = 0.64) but this relationship was moderate and negative in the KAATSU RT group (r = −0.57, *p* = 0.07). 

Changes in walking speed after 3 months correlated strongly and negatively with baseline walking speed in the control (r = −0.85, *p* = 0.002) and KAATSU RT group (r = −0.71, *p* = 0.015).

Changes in knee extensor strength after 3 months correlated strongly and negatively with baseline knee extensor strength in the control (r = −0.63, *p* = 0.05) but not in the KAATSU RT group (r = −0.18, *p* = 0.59).

### 3.6. Side Effects

Patients in the KAATSU RT completed the RT program without side effects. During the study, there were no hospitalizations, no signs for de novo cardiac symptoms, or for an exacerbation of cardiac conditions. Patients did not complain of delayed-onset muscle soreness (DOMS). All patients received warfarin to reduce the risks for thrombosis after cardiac surgery and the prothrombin time-international normalized ratio (PT-INR) was controlled at the target range in two groups (control group: 1.7 ± 0.4; KAATSU RT group: 1.7 ± 0.3). There was no subcutaneous hemorrhage and petechiae. CPK, a marker of muscle injury, was at the standard range both groups (control: at baseline: 120.7 ± 55.7 IU/L, after 3 months: 79.6 ± 22.1 IU/L; KAATSU RT: at baseline: 96.7 ± 78.8 IU/L, after 3 months: 79.0 ± 42.9 IU/L). D-dimer, a marker of thrombosis, in KAATSU RT was over the standard value at baseline (2.9 ± 7.3 μg/mL) but around standard level at after 3 months (0.5 ± 0.2 μg/mL).

## 4. Discussion

The present study provides the first evidence showing that low-intensity KAATSU RT increases muscle size and strength without side effects in patients early after cardiac surgery. Thus, low-intensity KAATSU RT appears to be an effective and safe method to improve muscle strength and size as an adjuvant to standard aerobic cardiac rehabilitation program.

RT with moderate blood flow restriction and low loads might be a viable alternative to high-load RT. Indeed, here we found that low-intensity KAATSU RT substantially increased muscle strength and size in patients early after cardiac surgery. Previous studies reported that KAATSU RT was accompanied by adverse side effects such as dizziness, subcutaneous hemorrhage, petechial hemorrhage, drowsiness, numbness, nausea, and itchiness [21,22,23]. We started KAATSU RT with a low volume, very low intensity, and low repetition number (intensity: 5–10 kg for leg extension, 20–30 kg for leg press, number of sets: 1, total number of daily repetitions: 20) and increased these training parameters gradually (intensity: 20–30% of 1-RM, number of sets: 3, total number of daily repetitions: 90). All patients received warfarin therapy even though petechial hemorrhage did not occur during the training. Because of such careful monitoring and delivery of the program, we did not observe any of the adverse side effects reported previously in conjunction with KAATSU RT [21,22,23]. We suspect that the medical precautions helped facilitate hemostasis and minimize the risk for thrombus formation. While pulmonary embolism and thrombosis have never been reported during KAATSU RT in cardiac patients, as a precaution, we still decided to measured D-dimer, a marker of thrombosis, but found normal levels at after 3 months. These data might be compatible with previous studies showing that vascular occlusion alone stimulates fibrinolytic activity without the coagulation activity and clot formation [34]. Umbel et al. [35] also reported low-intensity blood flow restricted exercise was followed by DOMS in healthy subjects. However, in the present study, we observed no DOMS, and CPK, a marker of muscle injury, was also at normal levels after 3 months of KAATSU RT, but we did not ask participants to report muscle soreness subjectively. Thus, well-monitored and stepwise-increased volume of KAATSU RT appears to be a safe exercise intervention to increase muscle strength and size in patients early after open cardiac surgery. Indeed, we conducted the present study over the 3 months and observed no major side effects. Several studies reported that blood flow restriction might induce a sympathoexcitatory pressor reflex originating from skeletal muscle and also cardiovascular complications evoked by the exercise pressor reflex [36,37]. Such side effects cannot be excluded, and future studies will be needed to provide assurances that KAATSU RT is safe in long-term training.

Low-intensity, short-term KAATSU RT can increase muscle strength and mass in athletes and healthy adults [12,13,14,15,16]. During conventional high-load RT, small, slow-type muscle fibers become activated according to the size principle [38]. With an increase in RT intensity, large and fast-type muscle fibers become also activated. However, during the KAATSU RT, the moderate blood flow restriction creates a hypoxic stimulus which is known to activate both small and large motor units, comprising, respectively, slow and fast muscle fibers [12,17,18,19]. Recently, we also observed that low-intensity KAATSU RT increased muscle activation during knee extension exercise in cardiovascular patients including those who have had open cardiac surgery [20]. In agreement with these previous data, here we observed that low-intensity KAATSU RT increased muscle strength and size in patients early after cardiac surgery. KAATSU RT induced greater increases in knee extensor strength (~37%) than muscle mass as measured by MTH (~20%), indirectly suggesting that KAATSU RT also increased neural activation of muscles measured during maximal voluntary knee extension.

We noticed that MTH and SMI decreased early after cardiac surgery in both control and KAATSU RT groups, and then recovered to baseline level in the control, but it was larger than at baseline in the KAATSU RT group after 3 months (Table 3). The reason for the slight decrease in muscle mass early after cardiac surgery in both groups could be partly related to proteolysis of skeletal muscle induced by post-operative elevation of inflammatory cytokine production [39] and short-term bedrest. However, the atrophy was minimized because patients spontaneously recovered after cardiac surgery, indicated by their ability to walk 200 m. The favorable KAATSU effects were clear because walking speed and knee extensor strength increased in the KAATSU RT group accompanying increase of muscle size, while the control group only returned to baseline after 3 months. Had we been able to measure leg muscle strength and size at a higher frequency, we suspect, we would have found faster rate of muscle recovery in the KAATSU RT compared with the control group. 

Muscle size and physical function increased after 3 months in each participant in KAATSU RT group (Figure 2). Therefore, negative correlations in KAATSU RT group indicate that when patients received KAATSU RT in combination with standard cardiac rehabilitation, patients with low vs. high physical function at baseline were more prone to gain muscle size and physical function. Correlation coefficients between baseline and changes after 3 months were also negative in the control group (Figure 2), but the changes after 3 months tended to be positive for patients with low baseline physical function and negative for patients with high baseline physical function. Thus, standard cardiac rehabilitation was an insufficient exercise stimulus to recover the loss of physical function in patients with high baseline physical function. Taken together, adding KAATSU RT on standard cardiac rehabilitation may facilitate the recovery of muscle size and physical function.

Sarcopenia is associated with post-surgery complications, including prolonged recovery and poor quality of life [3,4]. In the present study, 5 and 4 out of 10 patients had low muscle mass and low physical function (low hand grip or walking speed) according to an international standard [32] in the control group. In this group, early after open cardiac surgery, the number of patients with low muscle mass and low physical function increased, respectively, to 8 and then decreased to 5 and 4 after 3 months. On the other hand, in the KAATSU RT group, 7 and 4 out of 10 patients had low muscle mass and low physical function. Similar to the control group, the number of patients with low muscle mass and low physical function, respectively, increased to 8 and 5 early after cardiac surgery, but decreased to 2 and 1 after 3 months. Thus, it is likely that low-load KAATSU RT can reduce the cardiac surgery-induced loss in muscle mass and physical function, a speculation future studies will have to examine in more detail.

There are several limitations in the present study. We used BIA to measure muscle volume. Patients, especially with heart failure, are typically overhydrated and often have other conditions that might cause errors in BIA-estimated muscle mass [38,40]. However, in the present study, ECW/TBW, reflecting as edema, did not significantly change during the study in both groups. Thus, the influence of the edema-related errors in BIA-estimated muscle were probably small. In addition, we examined a small number of patients early after cardiac open surgery, especially females. Therefore, the future randomized trials will include a large number of patients to clarify the clinical usefulness and safety of the KAATSU RT training in patients early after cardiac open surgery. This limitation is reflected by the non-significant changes in any of the functional outcomes even though the effect size clearly favored the intervention group. The intervention requires a highly specialized piece of equipment and patients’ medical supervision, conditions that might limit the use of KAATSU RT in some hospitals.

## 5. Conclusions

Low-intensity KAATSU training for 3 months can increase muscle strength and size in patients early after cardiovascular surgery. Low-intensity KAATSU RT as an adjuvant to standard cardiac rehabilitation appears to safely increase skeletal muscle strength and size in cardiovascular surgery patients.

## Figures and Tables

**Figure 1 jcm-10-00547-f001:**
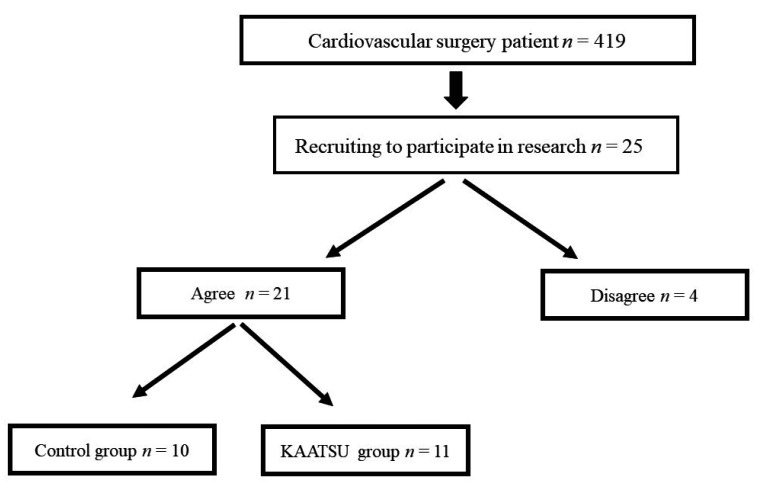
Patient selection.

**Figure 2 jcm-10-00547-f002:**
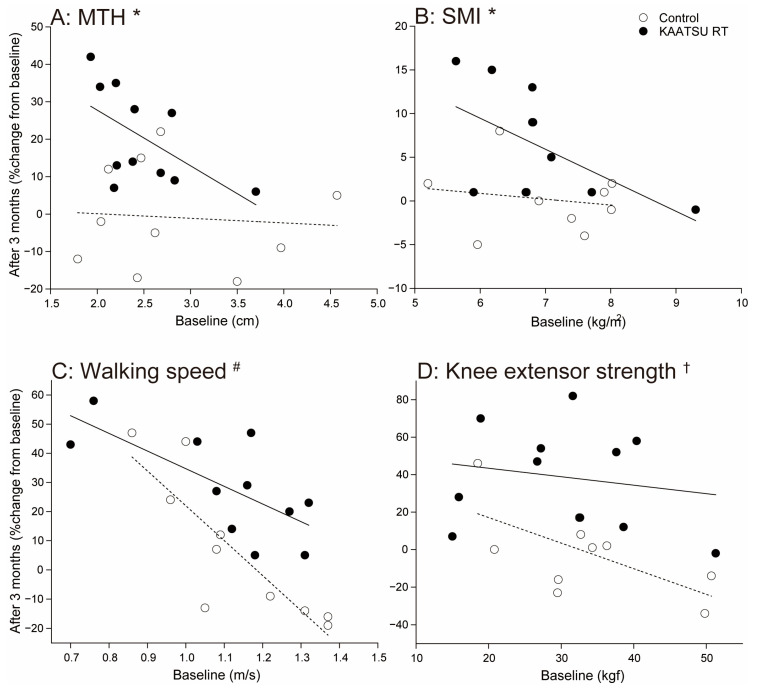
Correlations between baseline and changes after 3 months in MTH, SMI, walking speed, and knee extensor strength. A: correlation for MTH, B: correlation for SMI, C: correlation for walking speed, D: correlation for knee extensor strength. Open circles: the control group, filled circles: the moderate blood flow restriction (KAATSU RT) group. Dotted line: a regression line in the control group, continuous line: a regression line in the KAATSU RT group. *: Significant tendency in the KAATSU RT group (*p* < 0.1), ^#^: Significance in the control and the KAATSU RT group (*p* < 0.05), ^†^: Significance in the control group (*p* < 0.05).

**Table 1 jcm-10-00547-t001:** Inclusion and exclusion criteria.

Inclusion Criteria	Exclusion Criteria
# ability to perform the preoperative evaluation of leg extension strength # ability to perform postoperative cardiac rehabilitation program consisting of aerobic exercise # ability to provide written informed consent	# serious perioperative complications such as # pneumonia, instability of hemodynamics, heart failure, and cardiac arrhythmias # emergency surgery cases, dialysis patients, and patients who could not walk independently and perform resistance exercise # current neurological disorders or previous cerebral vascular accident with residual neurological deficit significant enough to limit exercise # malignant tumor, past fracture of the hip, pelvis, or femur, varicose veins, family or personal history of deep vein thrombosis, family or personal history of pulmonary embolism # patients with pacemaker implantation who cannot receive BIA methods

BIA, bioelectrical impedance analysis.

**Table 2 jcm-10-00547-t002:** The baseline patients’ characteristics.

Number	Control (*n* = 10)	KAATSU RT (*n* = 11)	*p* value	Cohen’s *d*
Male: Female	9: 1	9: 2		
Age, year	66 ± 8.7	57 ± 12.2	0.08	0.82
Body mass, kg	62.7 ± 8.3	61.5 ± 9.4	0.80	0.11
BMI, kg/m^2^	23.1 ± 2.3	22.1 ± 3.6	0.45	0.31
Height, cm	165.2 ± 7.4	167 ± 12	0.34	0.19
Risk factors, number				
Hypertension	7	8	0.63	
Diabetes	4	3	0.44	
Dyslipidemia	2	0	0.21	
Smoking	1	1	0.74	
Hemodialysis	0	0		
Atrial fibrillation	4	4	0.61	
NYHA I–IV, number	I: 0	I: 0		
	II: 9	II: 9	0.54	
	III: 1	III: 2	0.54	
	IV: 0	IV: 0		
Disease, number	Severe AS: 4	Severe AS: 3	0.44	
	Severe AR: 1	Severe AR: 4	0.18	
	Severe MS: 1	Severe MS: 1	0.74	
	Severe MR: 3	Severe MR: 3	0.63	
	A P: 1			
Cardiovascular surgery, number	AVR: 5	AVR: 6	0.59	
	MVR: 1	MVR: 1	0.74	
	MVP: 3	MVP: 3	0.63	
other	CABG: 1	Bentall: 1		
Creatinine, mg/dL	0.94 ± 0.2	1.06 ± 0.24	0.25	0.54
Hb, g/dL	13.6 ± 1.5	13.1 ± 2.0	0.51	0.31
Alb, g/dL	4.2 ± 0.2	3.9 ± 0.6	0.06	0.89
HbA1c, %	6.0 ± 0.4	5.8 ± 0.6	0.33	0.44
BNP, pg/ml	172 ± 138	303 ± 262	0.14	0.69
EF, %	59 ± 11	54 ± 16	0.31	0.45

The mean ± SD values are shown. BW, body mass; BMI, body mass index; NYHA, New York Heart Association; AS, aortic stenosis; AR, aortic regurgitation; MS, mitral stenosis; MR, mitral regurgitation; AP, angina pectoris; AVR, aortic valve replacement; MVR, mitral valve replacement; MVP, mitral valve plasty; CABG, coronary artery bypass grafting; Bentall, Bentall procedure; Hb, hemoglobin; Alb, albumin; HbA1c, hemoglobin A1c; BNP, Brain Natriuretic Peptide; EF, ejection fraction.

**Table 3 jcm-10-00547-t003:** Changes of morphological and physical function data in the two groups.

	Control			KAATSU RT		
	Baseline	Early after Cardiac Surgery	After 3 Months	Baseline	Early after Cardiac Surgery	After 3 Months
Body mass, kg	62.7 (8.3)	60.6 (8.2)	61.4 (8.0)	61.5 (9.4)	58.5 (7.9) *	62.2 (8.1) ^#^
BMI, kg/m^2^	23.1 (2.3)	22.2 (2.4)	24.1 (5.0)	22.1 (3.6)	20.9 (3)	23 (3.2)
Body fat volume, kg	17.5 (3.8)	16.4 (4)	16.9 (3.6)	16.4 (6.5)	14.9 (6)	16.4 (5.9)
Body fat percentage, %	25.4 (7.5)	27.1 (6.2)	25.9 (7)	25 (6.5)	25.4 (11)	25 (10)
ECW/TBW	0.39 (0.012)	0.39 (0.007)	0.40 (0.015)	0.39 (0.007)	0.40 (0.005)	0.39 (0.004)
MTH, cm	2.8 (0.9)	2.6 (1.0)	2.8 (0.9)	2.5 (0.5)	2.1 (0.4) *	3.0 (0.5) *^,#^
SMI, kg/m^2^	7.0 (0.9)	6.7 (0.9)	7.0 (1.0)	6.9 (1.0)	6.5 (0.8) *	7.3 (0.8) *^,#^
Walking speed, m/s	1.1 (0.2)	1.0 (0.2)	1.2 (0.1)	1.1 (0.2)	1.2 (0.2)	1.39 (0.2) *^,#^
Handgrip, kgf	31.3 (7.4)	28.3 (8.2)	30.7 (6.7)	30.3 (7.5)	29.2 (5.2)	33.9 (8.5)
Knee extensor, kgf	33.5 (10.5)	28 (10.4)	31.7 (7.48)	30.5 (11.2)	29.2 (5.2)	41.8 (15.1) *^,#^

The mean ± SD values are shown. * *p* < 0.05 vs. baseline, ^#^
*p* < 0.05 vs. early after cardiac surgery. BMI, body mass index; MTH, anterior mid-thigh wall thickness; SMI, skeletal muscle mass index; ECW, extracellular water; TBW, total body water.

## Data Availability

The data presented in this study are available on request from the corresponding author. The data are not publicly available due to ethical reason.

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
