# Peer review of "Low-Intensity Resistance Training with Moderate Blood Flow Restriction Appears Safe and Increases Skeletal Muscle Strength and Size in Cardiovascular Surgery Patients: A Pilot Study"

_jcm, 2021, doi:10.3390/jcm10030547_

Round 1

Reviewer 1 Report

This is an interesting and timely study examining the effects of low-load resistance training with blood flow restriction (BFR), so-called ‘Kaatsu training’ on skeletal muscle adaptations and functional performance in cardiac surgery patients. Importantly, great care and attention to the progression of training load and side/effects/safety was made – as one would expect for a preliminary study of this nature.

Overall, BFR resulted in positive adaptations in this clinical population, and it was considered safe and effective for cardiac rehabilitation, although clearly more work is required in a larger and broader clinical population, using perhaps more sensitive measurements of muscle mass etc.

The manuscript is reasonably well written although rather long. I have a few comments that might help provide some clarity in a few key places

Introduction

Line 62: please specify the typical low intensity e.g. 30% 1RM

Line 67. The authors are correct to address the side effects around hemorrhage, rhabdomyolysis etc. however, it might also be pertinent to consider the issue around cardiovascular complications e.g. engagement of the exercise pressor reflex, which contributes to the autonomic cardiovascular response to exercise that may increase the risk for deleterious cardiovascular events (Spranger MD, Krishnan AC, Levy PD, O'Leary DS, Smith SA. Blood flow restriction training and the exercise pressor reflex: a call for concern. Am J Physiol Heart Circ Physiol. 2015;309(9):H1440-52; Cristina-Oliveira M, Meireles K, Spranger MD, O'Leary DS, Roschel H, Peçanha T. Clinical safety of blood flow-restricted training? A comprehensive review of altered muscle metaboreflex in cardiovascular disease during ischemic exercise. Am J Physiol Heart Circ Physiol. 2020;318(1):H90-109). It may be worth mentioning here or at least addressing this issue in the discussion.

There should be a hypothesis!

Methods

Table 2. Body mass, not weight!

Muscle size measurement. Please provide the day-to-day variability in the ultrasound measurement of size (e.g. coefficient of variation), as this technique can be quite variable.

I would question the inclusion of handgrip strength. Although there may be some cross-transfer, the training regime was on the lower legs. But I do appreciate that handgrip is sometimes used as a global measure of “strength”.

Results

Table 3 is awkward to navigate. Perhaps have the baseline, early and 3 month measures adjacent to each other so the changes within groups can be more easily observed.

Where there are key significant changes, perhaps provide a % change within the text or table alongside the absolute values, to emphasize these changes more.

Discussion

This is quite long and may help by being condensed a little.

Not sure of the need for paragraph (lines 292-300) as this just restates the Introduction.

Line 329-338. Not sure how relevant the mechanistic discussion on motor unit recruitment or hypoxic stimulus is here, as it seems beyond the scope of the study. Perhaps focus on the functional observations e.g. whether the changes in strength translate to other function performance characteristics, as there are more important to patients.

Author Response

//

Reply to Reviewer #1

We greatly appreciate your careful attention to our manuscript and especially your excellent suggestions for improving the clarity and correctness of the message. We have corrected the paper as per your suggestions, and consider the revised manuscript much improved.

This is an interesting and timely study examining the effects of low-load resistance training with blood flow restriction (BFR), so-called ‘Kaatsu training’ on skeletal muscle adaptations and functional performance in cardiac surgery patients. Importantly, great care and attention to the progression of training load and side/effects/safety was made – as one would expect for a preliminary study of this nature.

Overall, BFR resulted in positive adaptations in this clinical population, and it was considered safe and effective for cardiac rehabilitation, although clearly more work is required in a larger and broader clinical population, using perhaps more sensitive measurements of muscle mass etc.

The manuscript is reasonably well written although rather long. I have a few comments that might help provide some clarity in a few key places

Introduction

Line 62: please specify the typical low intensity e.g. 30% 1RM

#) Answer: I added 20-30% of 1RM.

Page 2 line 64 through short-term low-intensity RT (20-30% of 1-RM) [12-16]

Line 67. The authors are correct to address the side effects around hemorrhage, rhabdomyolysis etc. however, it might also be pertinent to consider the issue around cardiovascular complications e.g. engagement of the exercise pressor reflex, which contributes to the autonomic cardiovascular response to exercise that may increase the risk for deleterious cardiovascular events (Spranger MD, Krishnan AC, Levy PD, O'Leary DS, Smith SA. Blood flow restriction training and the exercise pressor reflex: a call for concern. Am J Physiol Heart Circ Physiol. 2015;309(9):H1440-52; Cristina-Oliveira M, Meireles K, Spranger MD, O'Leary DS, Roschel H, Peçanha T. Clinical safety of blood flow-restricted training? A comprehensive review of altered muscle metaboreflex in cardiovascular disease during ischemic exercise. Am J Physiol Heart Circ Physiol. 2020;318(1):H90-109). It may be worth mentioning here or at least addressing this issue in the discussion. however, it might also be pertinent to consider the issue around cardiovascular complications e.g. engagement of the exercise pressor reflex, which contributes to the autonomic cardiovascular response to exercise that may increase the risk for deleterious cardiovascular events (Spranger MD, Krishnan AC, Levy PD, O'Leary DS, Smith SA. Blood flow restriction training and the exercise pressor reflex: a call for concern. Am J Physiol Heart Circ Physiol. 2015;309(9):H1440-52; Cristina-Oliveira M, Meireles K, Spranger MD, O'Leary DS, Roschel H, Peçanha T. Clinical safety of blood flow-restricted training? A comprehensive review of altered muscle metaboreflex in cardiovascular disease during ischemic exercise. Am J Physiol Heart Circ Physiol. 2020;318(1):H90-109). It may be worth mentioning here or at least addressing this issue in the discussion.

#) Answer: Thank you very much for your suggestions. We have added the following sentences into the discussion and cited two papers.

Page 9 line 340: Indeed, we conducted the present study over the 3 months and observed no major side effects. Several studies reported that blood flow restriction might induce a sympatho-excitatory pressor reflex originating from skeletal muscle and also cardiovascular com-plications evoked by the exercise pressor reflex [36, 37]. Such side effects cannot be ex-cluded, and future studies will be needed to provide assurances that KAATSU RT is safe in long-term training.

I cited two papers.

  1. Spranger, M.D.; Krishnan, A.C.; Levy, P.D.; O'Leary, D.S.; Smith, S.A. Blood flow restriction training and the exercise pressor reflex: a call for concern. J. Physiol. Heart Circ. Physiol. 2015, 309, H1440-1452.
  2. Cristina-Oliveira M.; Meireles, K.; Spranger, M.D.; O'Leary, D.S.; Roschel, H.; Peçanha, T. Clinical safety of blood flow-restricted training? A comprehensive review of altered muscle metaboreflex in cardiovascular disease during ischemic exercise. J. Physiol. Heart Circ. Physiol. 2020, 318, H90-109.

There should be a hypothesis!

#) Answer: We have added a hypothesis at the bottom of the Introduction section.

Methods

Table 2. Body mass, not weight!

#) Answer: I corrected it.

Muscle size measurement. Please provide the day-to-day variability in the ultrasound measurement of size (e.g. coefficient of variation), as this technique can be quite variable.

#) Answer: You are right. Ink markers on the anterior and posterior thigh and posterior lower leg were used to ensure similar positioning over repeated MTH measurements. And, I showed the coefficient of variation in the ultrasound measurements. 

Page 5 line 193: Ink markers on the anterior and posterior thigh and posterior lower leg were used to ensure similar positioning over repeated MTH measurements. The measurement was performed twice on the right thigh and the average of the two values were used in the analysis. The test-retest reliability (Intraclass Correlation Coefficient (ICC), standard error of measurement (SEM) and minimal difference) was previously determined using the data of 9 older women measured twice within few days for MTH (0.994, 0.28 cm, 0.79 cm).                     

I would question the inclusion of handgrip strength. Although there may be some cross-transfer, the training regime was on the lower legs. But I do appreciate that handgrip is sometimes used as a global measure of “strength”.

#) Answer: You are right. We used the criteria of muscle weakness as sarcopenia, so need to measure the handgrip strength.

Results

Table 3 is awkward to navigate. Perhaps have the baseline, early and 3 month measures adjacent to each other so the changes within groups can be more easily observed.

#) Answer: I have revised Table 3 according to your comment.

Where there are key significant changes, perhaps provide a % change within the text or table alongside the absolute values, to emphasize these changes more.

#) Answer: Thank you so much for your helpful advice. I have added the delta and % change relative to the baseline in the text and revised sentences.

Discussion

This is quite long and may help by being condensed a little.

#) Answer: I arranged the discussion.

Not sure of the need for paragraph (lines 292-300) as this just restates the Introduction.

#) Answer: I deleted this part.

Line 329-338. Not sure how relevant the mechanistic discussion on motor unit recruitment or hypoxic stimulus is here, as it seems beyond the scope of the study. Perhaps focus on the functional observations e.g. whether the changes in strength translate to other function performance characteristics, as there are more important to patients.

 #) Answer: I agree with you. I deleted some part. However, we had previously reported that KAATSU RT can enhance muscle activation during low-intensity knee extension exercise in cardiovascular patients with or without cardiovascular surgery [20]. Therefore, we partly mentioned about it.

Reviewer 2 Report

This is an important and well structured study. It begins to fill an important gap in practice for cardiac rehabilitation by immediately reducing effects on muscle atrophy that occur post-surgery. This is something that should not be understated.  

However, there are omissions/issues in your manuscript that I believe, when corrected, would strongly support and improve your potential publication.

It appears that neither your BFR or Control group were provided the standard practice for cardiac rehabilitation of an aerobic exercise program. In a methodological and ethical sense, this seems a strange choice. There are clear benefits to this form of rehabilitation that participants in both groups could have received, though as part of this study, volunteers either received a BFR resistance protocol, which has not yet been justified to provide the same basic health benefits, or no cardiac rehabilitation (Control) for the duration of the study. This means that by volunteering for the study, participants (especially those in the Control group) were worse off than they would have been in a standard care scenario. All participants in the study were cleared as having 'an ability to perform [a] postoperative cardiac rehabilitation program consisting of aerobic exercise' so why was this not the case? What was the reasoning for not including the BFR protocol in addition to standard care?

Regardless, your discussion/conclusions should be changed to pose BFR training as a potential addition to a standard aerobic cardiac rehabilitation program. It would not be appropriate for this to be recommended in isolation.  

Some more transparency would also benefit this study. Final sample numbers are provided. However, how many potential subjects were identified and approached? How many agreed? How many were screened out of the program? Were there drop-outs? This could be presented in a figure. It would also increase the assessed quality of your study in future peer reviews. 

My final major concern is that no group differences were identified despite interactions occurring. This suggests the study being underpowered, at least for some measures. However, this was never directly addressed in the discussion. Did you power for a specific measure? All measures? Either way, these results and the somewhat low numbers for a clinical trial suggest the necessity to reframe this study as a pilot.

Some specific notes:

  • Title: This is very conclusive. It is likely more appropriate to frame the training as 'appears safe' rather than 'is safe'. 
  • Line 66: The risks of BFR are reported on three occasions. This should be refined. The notes on rhabdomyolysis are also not necessary as it occurred rarely, and is not terribly relevant to this population. Instead further focus should be given to this population and the risks of adverse events that they may be predisposed to.
  • Line 83: Who assessed/defined if participants met the selection criteria?
  • Line 84: Grammar.
  • Line 123: Was this the duration for the repetitions or for a specific phase (i.e. eccentric)?
  • Line 129: What source was used to determine the 1-RM via the 8 or 10-RM?
  • Line 140: This comment is based more-so on overall design. Why choose handgrip strength as a priority measure? It is representative of physical function. However, there would be other tests more appropriate to the training performed (i.e. timed up-and-go, 6-min walk test, sit-to-stand)
  • Line 161: BIA is later discussed as a limitation. However, did you take steps to ensure some consistency in this measure? e.g. hydration status etc. If so, please report.
  • Line 200: Grammar.
  • Table 3: This likely isn't possible considering table size, but reporting change scores would be beneficial.
  • Line 221: While the side effects are important to report, I don't think they should be presented immediately after an unrelated table.
  • Results: A couple of quick notes. Why weren't main effects reported, at least in the table? Also, there is a habit in this section to often state that there were increases/decreases when this didn't was not supported by statistical analysis (p > 0.05). These are not changes and shouldn't be stated as so. Values remained similar.
  • Line 268: I understand the reasoning for including correlational analyses but I'm not sure that, as currently presented, this data provides much to explain changes within your groups. This is most reflected in your discussion of these results. It is stated that those in the Control group who had high baseline physical function were 'more prone to losing physical function'. Whereas, those lower functioning participants at baseline in the BFR group were more 'prone to gain muscle size'. However, in terms of just a correlation statistic, these are different interpretations for the same effect. I would suggest either removing these analyses or consider adding a correlation figure/figures to present the data in a way that supports your conclusions made.
    Also please note that in section 3.6, it can be unclear what group the data refers to in the separate paragraphs. 
  • Line 306: From here, you list numerous safety precautions taken, though many of which should probably have been presented in your methods instead.
  • Line 324: Please add that, though DOMS wasn't observed, it may have been if assessed subjectively as well.
  • Line 341: This is a poor conclusion to make. Early adaptations to a training program are always going to be more-so neuromuscular rather than hypertrophic. A direct comparison of % change is also not appropriate because it is an arbitrary comparison.
  • Line 350: 'Started to spontaneously recover'. Please change this terminology.
  • Line 356: This is a strong conclusion to make. However, before doing so, you need to consider what the minimal important differences are in changes to muscle strength and thickness for functional beneficial affects to the patient.
  • Line 380: This needs to be explained further. What was the reasoning for doing so? What effects could this have had?
  • Conclusion: Your conclusion should only go as far as to be recommend as  BFR training as a likely safe inclusion to cardiac rehabilitation programs in addition to standard practice. 

Author Response

//

Reply to Reviewer #2

We greatly appreciate your careful attention to our manuscript and especially your excellent suggestions for improving the clarity and correctness of the message. We have corrected the paper as per your suggestions, and consider the revised manuscript much improved.

This is an important and well-structured study. It begins to fill an important gap in practice for cardiac rehabilitation by immediately reducing effects on muscle atrophy that occur post-surgery. This is something that should not be understated. However, there are omissions/issues in your manuscript that I believe, when corrected, would strongly support and improve your potential publication.

It appears that neither your BFR or Control group were provided the standard practice for cardiac rehabilitation of an aerobic exercise program. In a methodological and ethical sense, this seems a strange choice. There are clear benefits to this form of rehabilitation that participants in both groups could have received, though as part of this study, volunteers either received a BFR resistance protocol, which has not yet been justified to provide the same basic health benefits, or no cardiac rehabilitation (Control) for the duration of the study. This means that by volunteering for the study, participants (especially those in the Control group) were worse off than they would have been in a standard care scenario. All participants in the study were cleared as having 'an ability to perform [a] postoperative cardiac rehabilitation program consisting of aerobic exercise' so why was this not the case? What was the reasoning for not including the BFR protocol in addition to standard care?

#) Answer: I agree with you. All patients had received a standard inpatient cardiac rehabilitation program consisting of aerobic exercise. We mentioned about it in methods.

Page 3 line 99: All patients had received the inpatients standard cardiac rehabilitation program consisting of 30 minutes aerobic exercise within intensity of anerobic threshold (AT) level on a cycle ergometer. After discharge, patients in the control group performed the standard cardiac rehabilitation program in the hospital twice a week for 3 months but did not receive KAATSU RT. Patients in KAATSU RT group were joined the KAATSU RT program twice a week adding on standard cardiac rehabilitation in the hospital for 3 months.

Regardless, your discussion/conclusions should be changed to pose BFR training as a potential addition to a standard aerobic cardiac rehabilitation program. It would not be appropriate for this to be recommended in isolation.  

#) Answer: I agree with you. I changed the conclusions.

Low-intensity KAATSU RT as an adjuvant to standard cardiac rehabilitation appears to safely increase skeletal muscle strength and size in cardiovascular surgery patients.

Some more transparency would also benefit this study. Final sample numbers are provided. However, how many potential subjects were identified and approached? How many agreed? How many were screened out of the program? Were there drop-outs? This could be presented in a figure. It would also increase the assessed quality of your study in future peer reviews. 

#) Answer: Thank you very much for your suggestion. I summarized patient selection and represented in Figure 1.

Page 2 line 83: Of the inpatients receiving cardiac open surgery between April 2017 and June 2020, a total of 25 patients, who met the following criteria (Table 1), were recruited to participate the present study (Figure 1).

My final major concern is that no group differences were identified despite interactions occurring. This suggests the study being underpowered, at least for some measures. However, this was never directly addressed in the discussion. Did you power for a specific measure? All measures? Either way, these results and the somewhat low numbers for a clinical trial suggest the necessity to reframe this study as a pilot.

#) Answer: I agree with your comments. We did not measure power for a specific measure. We added a pilot study in the title.

Low-intensity Resistance Training with Moderate Blood Flow Restriction Appears Safe and Increases Skeletal Muscle Strength and Size in Cardiovascular Surgery Patients: A Pilot Study

Some specific notes:

  • Title: This is very conclusive. It is likely more appropriate to frame the training as 'appears safe' rather than 'is safe'. 

#) Answer: I changed it.

  • Line 66: The risks of BFR are reported on three occasions. This should be refined. The notes on rhabdomyolysis are also not necessary as it occurred rarely, and is not terribly relevant to this population. Instead further focus should be given to this population and the risks of adverse events that they may be predisposed to.

#) Answer: I deleted some parts, and refined the risks.

  • Line 83: Who assessed/defined if participants met the selection criteria?

#) Answer: Patients’ surgeons determined if a patient met the inclusion criteria.

  • Line 84: Grammar.

#) Answer: I changed it as follows.

Page 2 line 83: In inpatients receiving cardiac open surgery between April 2017 and June 2020, total 25 patients, who met the following criteria (Table 1), recruited to participate the present study as shown in Figure 1.

  • Line 123: Was this the duration for the repetitions or for a specific phase (i.e. eccentric)?

#) Answer: I added the following sentences.

Page 4 line 133: Paced by a metronome, the duration of the specific phase (eccentric, and concentric) was 1.5s for both exercises.

  • Line 129: What source was used to determine the 1-RM via the 8 or 10-RM?

#) Answer: We usually used this method to measure 1-RM according to ACSM’s recommendation based on following 2 references.

Fleck SJ and Kraemer WJ. Designing Resistance Training Programs. 4th ed. Champaign, IL: Human Kinetics, 1-62, 179-296, 2014.

Stone MH and O’Bryant HS. Weight Training: A Scientific Approach. Minneapolis: Burgess, 104-190, 1987.

  • Line 140: This comment is based more-so on overall design. Why choose handgrip strength as a priority measure? It is representative of physical function. However, there would be other tests more appropriate to the training performed (i.e. timed up-and-go, 6-min walk test, sit-to-stand)

#) Answer: You are right. But we used the criteria of muscle weakness as sarcopenia, so needed to measure the handgrip strength.

  • Line 161: BIA is later discussed as a limitation. However, did you take steps to ensure some consistency in this measure? e.g. hydration status etc. If so, please report.

#) Answer: I added the following sentences.

Page 10 line 391: However, in the present study, ECW/TBW, reflecting as edema, did not significantly change during the study in both groups. Thus, the influence of the edema-related errors in BIA-estimated muscle were probably small.

  • Line 200: Grammar.

#) Answer: I corrected the sentences.

Page 6 line 214: We computed Pearson’s correlation coefficient to determine if baseline muscle size and function would be associated with changes in muscle size and function after 3 months.

  • Table 3: This likely isn't possible considering table size, but reporting change scores would be beneficial.

#) Answer: According to your and the other reviewer’s comments, we have rearranged Table 3 and revised sentences in the Results section.

  • Line 221: While the side effects are important to report, I don't think they should be presented immediately after an unrelated table.

#) Answer: I moved the side effects section to the last part of the results.

  • Results: A couple of quick notes. Why weren't main effects reported, at least in the table? Also, there is a habit in this section to often state that there were increases/decreases when this didn't was not supported by statistical analysis (p > 0.05). These are not changes and shouldn't be stated as so. Values remained similar.

#) Answer: Based on your comment, we have included the time main effect in BMI, body fat volume, and handgrip strength. You also have pointed, number of patients was somewhat low in this study thus p values did not reach the significant level but effect size reached meaningful level (d>0.2). Then, we reported these values as not significant but slightly/weakly changed. But we have toned down these values.

  • Line 268: I understand the reasoning for including correlational analyses but I'm not sure that, as currently presented, this data provides much to explain changes within your groups. This is most reflected in your discussion of these results. It is stated that those in the Control group who had high baseline physical function were 'more prone to losing physical function'. Whereas, those lower functioning participants at baseline in the BFR group were more 'prone to gain muscle size'. However, in terms of just a correlation statistic, these are different interpretations for the same effect. I would suggest either removing these analyses or consider adding a correlation figure/figures to present the data in a way that supports your conclusions made.
    Also please note that in section 3.6, it can be unclear what group the data refers to in the separate paragraphs. 

#) Answer: We thank you for noticing and finding important results. We have made figures between baseline and the changes after 3 months relative to baseline, and found that, after all, patients with high baseline physical function were more prone to losing physical function and lower functioning patients at baseline were more prone to gain muscle size. We have added Figure 2 and revised discussion.

  • Line 306: From here, you list numerous safety precautions taken, though many of which should probably have been presented in your methods instead.

#) Answer: I removed it in methods.

  • Line 324: Please add that, though DOMS wasn't observed, it may have been if assessed subjectively as well.

#) Answer: However, in the present study, we observed no delayed-onset muscle soreness, and CPK, a marker of muscle injury, was also at normal levels after 3 months of KAATSU RT. We added the following sentence.

However, in the present study, we observed no DOMS, and CPK, a marker of muscle injury, was also at normal levels after 3 months of KAATSU RT but we did not ask participants to report subjectively muscle soreness.

  • Line 341: This is a poor conclusion to make. Early adaptations to a training program are always going to be more-so neuromuscular rather than hypertrophic. A direct comparison of % change is also not appropriate because it is an arbitrary comparison.

#) Answer: We have carefully revised this sentence.

  • Line 350: 'Started to spontaneously recover'. Please change this terminology.

#) Answer: I changed the following sentence.

However, the atrophy was minimized because patients spontaneously recovered after cardiac surgery, indicated by their ability to walk 200m.

  • Line 356: This is a strong conclusion to make. However, before doing so, you need to consider what the minimal important differences are in changes to muscle strength and thickness for functional beneficial affects to the patient.

#) Answer: We have carefully revised the discussion for correlation analysis based on newly added Figure 2.

  • Line 380: This needs to be explained further. What was the reasoning for doing so? What effects could this have had?

#) Answer: I added the following sentence.

Page 9 line 399: However, in the present study, ECW/TBW, reflecting as edema, did not significantly change during the study in both groups. Thus, the influence of the edema-related errors in BIA-estimated muscle were probably small.

  • Conclusion: Your conclusion should only go as far as to be recommend as BFR training as a likely safe inclusion to cardiac rehabilitation programs in addition to standard practice.

#) Answer: I agree with you. The discussion/conclusions have been changed to pose BFR training as a potential addition to a standard aerobic cardiac rehabilitation program.

Round 2

Reviewer 2 Report

Well done with your consideration of my comments and the rapid edits to the manuscript. It now reads in a more transparent form and removes my ethical concerns.

I have a few quick comments that I think should be addressed:

  • Some edits you have made have incurred grammatical errors in the work. Please look over it again.
  • The units for the y-axis in your correlation figures (% Baseline) are unclear. Is this representing a % change from baseline? Currently it sounds like the y-axis values are are representative of a percentage of baseline values, despite that not being the case.
  • In the results, the strength of correlations should be interpreted as per standard norms (i.e. weak, strong etc.)
  • My main issue is how you continue to report results. 
    e.g. "Compared with baseline, body mass tended to decrease early after cardiac surgery in the control group (p=0.061, d=0.88, -2.1 kg, -3.3%) and then slightly increased after 3 months but was still slightly below baseline in this group (p=0.11, d=0.78, -1.3 kg, -2.1%)."
    I have no issue with reporting trends. However, consistently terms like 'increased' or 'was still slightly below baseline' are inappropriate. The mean values may not be the same but to state that they are below baseline is not true if p > 0.05. The body mass at those time-points were 'similar'. 

Author Response

///

Re//

Reply to Reviewer #2

We greatly appreciate your careful attention to our manuscript and especially your excellent suggestions for improving the clarity and correctness of the message. We have corrected the paper as per your suggestions, and consider the revised manuscript much improved.

Well done with your consideration of my comments and the rapid edits to the manuscript. It now reads in a more transparent form and removes my ethical concerns.

I have a few quick comments that I think should be addressed:

  • Some edits you have made have incurred grammatical errors in the work. Please look over it again.

#) Answer: Thank you very much for your suggestions. We have carefully checked and revised whole of the manuscript.

  • The units for the y-axis in your correlation figures (% Baseline) are unclear. Is this representing a % change from baseline? Currently it sounds like the y-axis values are representative of a percentage of baseline values, despite that not being the case.

#) Answer: We have renamed the y-axis unit “% Baseline” into “%change from baseline”.

  • In the results, the strength of correlations should be interpreted as per standard norms (i.e. weak, strong etc.)

#) Answer: We have added the strength of correlation and revised the sentences in the correlation analyses section.

  • My main issue is how you continue to report results.

e.g. "Compared with baseline, body mass tended to decrease early after cardiac surgery in the control group (p=0.061, d=0.88, -2.1 kg, -3.3%) and then slightly increased after 3 months but was still slightly below baseline in this group (p=0.11, d=0.78, -1.3 kg, -2.1%)."

I have no issue with reporting trends. However, consistently terms like 'increased' or 'was still slightly below baseline' are inappropriate. The mean values may not be the same but to state that they are below baseline is not true if p > 0.05. The body mass at those time-points were 'similar'.

#) Answer: We fully agreed with your comments. Therefore, we have tone down for reporting trends in Results section.
